# ALL BY LARGE LANGUAGE MODEL ITSELF

## ABSTRACT

The scaling laws constitute one of the fundamental principles of large language models (LLMs), which reveal that the model performance constantly improves as the training data increase. In this paper, we propose dynamic reinforcement learning (RL), which takes a step to achieve the scalability of RL for training the LLM by itself. Dynamic RL operates by sampling data from the dynamically changed LLM itself, estimating golden answers based on the model's own outputs, and then using this self-generated data to optimize the model. Its dynamic characteristic allows the data distribution to continuously adapt to the evolving model, leading to better alignment between training data and model capabilities. Unlike conventional approaches, dynamic RL requires neither static, pre-collected datasets nor external verifiers for correctness. All is done by the large language model itself. Experimental results demonstrate that dynamic RL can continually improve model performance over a thousand of training steps and achieve results comparable to models trained on large-scale external datasets.

## 1 INTRODUCTION

Large language models (LLMs) have demonstrated remarkable performance across a wide range of tasks (Jaech et al., 2024; Guo et al., 2025). A key factor driving this success is the principle of scaling laws (Kaplan et al., 2020), which shows that LLM performance improves as the amount of training data increases. LLMs can be trained using different learning paradigms, such as supervised learning and reinforcement learning (RL), each exhibiting different scalability characteristics.

In supervised learning, models are trained on data sampled from a static distribution, requiring pre-collected datasets of questions paired with human-labeled solutions. However, these human-labeled solutions are costly and finite, which limits scalability. In contrast, DeepSeek-R1 (Guo et al., 2025) demonstrates that RL can train models without relying on human-labeled solutions by sampling solutions from the dynamic LLM itself, thereby achieving great scalability. Since the questions are sampled from a static distribution while the solutions are sampled from a dynamic distribution, we refer to this paradigm as *semi-dynamic reinforcement learning*.

Despite its advantages, semi-dynamic RL still depends on static datasets of human-created questions and human-labeled answers, which remain finite and costly. Moreover, since the questions are sampled from a static distribution that may not align with the evolving LLM, the number of effective training questions gradually decreases over time (Yu et al., 2025; Zheng et al., 2025). This mismatch between static data and a constantly evolving model ultimately constrains scalability.

To address this limitation, we propose *dynamic reinforcement learning*, which takes a step toward achieving scalable RL using the LLM itself. In this framework, the LLM autonomously generates both questions and solutions, learning directly from its own self-sampled data without relying on external datasets or verifiers for correctness. All is done by the large language model itself. The dynamic nature of this approach allows the data distribution to evolve continuously alongside the model, ensuring better alignment between training data and model capabilities.

Transitioning from semi-dynamic RL to dynamic RL introduces additional challenges, most notably the absence of golden answers and the risk of mode collapse. Since the questions are generated by the LLM itself, human-labeled golden answers are unavailable and should instead be estimated. The core design principle of our dynamic RL is to encourage the model to generate relatively more questions whose answers can be reliably estimated by the estimation method, rather than to devise a new estimation method.

To realize this principle, we adopt three strategies. First, we estimate golden answers using majority voting (Wang et al.) and introduce a question reward function that promotes the generation of questions suitable for this estimation. Second, we design prompts and introduce filtering rules to exclude questions that are likely to be incorrectly answered. Third, we adjust the training dynamics by tuning hyperparameters so that the model produces questions of moderate difficulty that match with the estimation method.

Another major challenge is mode collapse, a phenomenon in which generated data degenerates into a limited set of modes (Kossale et al., 2022). In dynamic RL, this manifests as question collapse, where the model repeatedly produces similar questions, and answer collapse, where the model defaults to identical answers. Such collapse ultimately leads to performance degradation. For example, the model may repeatedly generate nearly identical questions such as "Solve the equation $2x + 3 = 7$", or consistently output the same answer, such as "\boxed{1}", across different questions.

To mitigate this issue, we introduce a diversity reward function that prompts the model to generate new questions with diverse estimated golden answers. This mechanism effectively enhances both question and answer diversity, thereby alleviating mode collapse.

Finally, we experimentally validate the scalability of dynamic RL. Our results demonstrate that dynamic RL can enhance model performance over a thousand of training steps by itself, and achieve accuracy comparable to that of semi-dynamic RL trained on large-scale, pre-collected static datasets.

## 2 DYNAMIC REINFORCEMENT LEARNING

**Learning Scalability** Scaling laws represent one of the fundamental laws in LLMs, which show that model performance can improve as model size, dataset size, and compute scale up. This phenomenon underpins the success of LLMs, as it suggests that model capability scales with resources.

We discuss the learning scalability in terms of training paradigms. LLMs can be trained through unsupervised learning, supervised learning, and RL. The supervised and unsupervised learning objectives can be expressed as

$$\mathcal{J}_{sl}(\theta) = \mathbb{E}_{q \sim \phi, \, s \sim \mu(\cdot|q)} \Big[ \log \pi_\theta(s|q) \Big],$$

where $\phi$ is the distribution over questions $q$, $\mu(\cdot|q)$ is the distribution of solutions $s$ conditioned on $q$, and $\pi_\theta$ is the LLM policy model parameterized by $\theta$. Unsupervised learning can be regarded as a special case where the conditioning variable $q$ is absent. Since both $q$ and $o$ are drawn from static distributions, we refer to this paradigm as *static learning*. However, the policy model $\pi_\theta$ constantly changes during training, while the data are sampled from a static distribution that can not adapt to the evolving policy model $\pi_\theta$. This mismatch inherently limits the scalability of learning.

RL, in contrast, optimizes a different objective:

$$\mathcal{J}_{rl}(\theta) = \mathbb{E}_{q \sim \phi, \, s \sim \pi_\theta(\cdot|q)} \Big[ R(q, s) \Big],$$

where $R(q, s)$ denotes the reward function evaluating the quality of a solution $s$ given a question $q$. Recent approaches, such as DeepSeek-R1, demonstrate that RL can enhance scalability by allowing the solutions sampled from the dynamic policy model to refine itself. Since the solutions $s$ are sampled from the evolving distribution $\pi_\theta(\cdot|q)$ during training, this paradigm can be described as *semi-dynamic reinforcement learning*. However, the questions $q$ remain drawn from a static distribution $\phi$, which continues to constrain scalability.

**Dynamic Reinforcement Learning** To further enhance scalability of RL, we propose a framework termed *dynamic reinforcement learning*. In this framework, we first sample questions from the policy model $\pi_\theta$ itself, then generates corresponding solutions from the policy model $\pi_\theta$ itself, and finally leverages the sampled data to optimize itself. All is done by the policy model $\pi_\theta$ itself. By continually sampling from dynamically evolving distributions $\pi_\theta$, the policy model can iteratively improve its performance. To this end, we optimize two objective functions: one for generating higher-quality solutions and another for generating higher-quality questions.

We first introduce the objective function $\mathcal{J}_s(\theta)$, which optimizes the quality of generated solutions:

$$\mathcal{J}_s(\theta) = \mathbb{E}_{q \sim \pi_\theta(\cdot|z),\, s \sim \pi_\theta(\cdot|q)} \Big[ R_s(q, s) \Big], \tag{1}$$

where $\pi_\theta(\cdot|z)$ denotes the distribution of questions $q$ given a prompt $z$, and $\pi_\theta(\cdot|q)$ denotes the distribution of solutions $s$ given a question $q$. The function $R_s(q, s)$ serves as the solution reward, evaluating the quality of the generated solutions $s$ for the corresponding question $q$.

Next, we define the objective function $\mathcal{J}_q(\theta)$, which optimizes the quality of generated questions:

$$\mathcal{J}_q(\theta) = \mathbb{E}_{z \sim p(z),\, q \sim \pi_\theta(\cdot|z)} \Big[ R_q(z, q) \Big], \tag{2}$$

where $R_q(z, q)$ is the question reward function, assessing the quality of a question $q$ given a prompt $z$. The distribution $p(z)$ specifies how prompts $z$ are sampled.

In summary, we jointly optimize $\mathcal{J}_s(\theta)$ and $\mathcal{J}_q(\theta)$ to enhance both the quality of solutions and the quality of questions. In the following, we detail the design of the prompt distribution $p(z)$, solution reward function $R_s(q, s)$ and the question reward function $R_q(z, q)$.

**Prompt**    Since this paper focuses on mathematical reasoning, we employ only a single type of prompt, denoted as $z_0$, with $p(z_0) = 1$, meaning that the prompt $z$ is always fixed to $z_0$. The prompt $z_0$ is explicitly designed to emphasize mathematical reasoning, as illustrated in the following box. It consists of three sentences: the first instructs the model to generate a single math question along with its solution. Since generating a question may also lead the model to implicitly produce a solution, we allow the model to output both the question and its solution directly. The second sentence aims to prohibit questions with non-unique answers, while the third sentence specifies the required output format, from which we retain only the question part.

> **Prompt $z_0$:** Generate exactly one math question and its step-by-step solution. The answer to the question should exist and be unique.
> Format the output as follows:
> Question: <math question here>
> Solution: <step-by-step solution here>

**Solution Reward Function**    To define the solution reward function $R_s(q, s)$, we first require a golden, or reference answer, for each question $q$. Since such answers are generally unavailable in dynamic RL, we estimate the golden answer using a majority-voting scheme (Wang et al.).

Specifically, for each question $q$, we sample $m$ solutions $\{s_j\}_{j=1}^m$ from the distribution $\pi_\theta(\cdot|q)$. For each solution $s_j$, we extract its final answer $a_j$ via a function $e(\cdot)$, which can be regular expressions or LLMs (Guo et al., 2025). We then define majority voting using a similarity metric $S_a(\cdot, \cdot)$ (Guo et al., 2025; Team et al., 2025). For any two answers $a_{j_1}$ and $a_{j_2}$, the similarity is given by

$$S_a(a_{j_1}, a_{j_2}) = \begin{cases} 1, & \text{if } a_{j_1} \text{ and } a_{j_2} \text{ are mathematically equivalent,} \\ 0, & \text{otherwise.} \end{cases} \tag{3}$$

The majority-voted golden answer $l(q)$ and its support size $r(q)$ are defined as

$$l(q) = \arg\max_{a \in \{a_j\}_{j=1}^m} \sum_{k=1}^m S_a(a, a_k), \tag{4}$$

$$r(q) = \max_{a \in \{a_j\}_{j=1}^m} \sum_{k=1}^m S_a(a, a_k), \tag{5}$$

where $l(q)$ represents the estimated golden answer for $q$, and $r(q)$ denotes the number of answers that are mathematically equivalent to $l(q)$.

Finally, the solution reward function $R_s(q, s)$ evaluates whether the extracted answer $e(s)$ matches the majority-voted golden answer $l(q)$:

$$R_s(q, s) = S_a\big(l(q), e(s)\big). \tag{6}$$

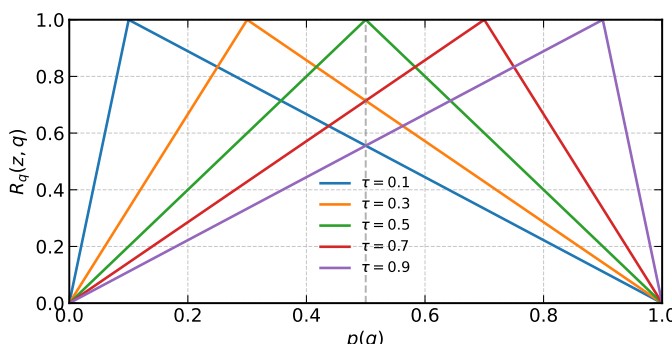

Figure 1: Question Reward function $R_q(z, q)$ with different $\tau$.

**Question Reward Function**   We define the question reward function based on the mean solution reward of a question $q$. For a given question $q$, the mean solution reward $p(q)$ is defined as

$$p(q) = \mathbb{E}_{s \sim \pi_\theta(\cdot|q)}[R_s(q, s)|q] \approx \text{Mean}(\{R_s(q, s_j)\}_{j=1}^m) = \frac{r(q)}{m}. \tag{7}$$

Here, $p(q)$ can be seen as an approximate measure of the difficulty of question $q$: larger values of $p(q)$ correspond to easier questions, while smaller values correspond to harder ones.

The question reward function $R_q(z, q)$ is then defined as

$$R_q(z, q) = \begin{cases} \dfrac{p(q)}{\tau}, & \text{if } 0 \le p(q) \le \tau, \\ \dfrac{1 - p(q)}{1 - \tau}, & \text{if } \tau < p(q) \le 1, \end{cases} \tag{8}$$

where $\tau \in [0, 1]$ is a hyperparameter. Figure 1 illustrates the behavior of $R_q(z, q)$ under different values of $\tau$. As $p(q)$ increases from 0 to $\tau$, the reward $R_q(z, q)$ increases; as $p(q)$ increases further from $\tau$ to 1, the reward decreases.

The question reward function $R_q(z, q)$ plays both a collaborative and an adversarial role with respect to the solution reward function $R_s(q, s)$. It is collaborative in that, when $p(q) \le \tau$, it drives the model to generate easier questions, thereby reinforcing the increase of $\mathcal{J}_s(\theta)$. At the same time, it is adversarial because, when $p(q) > \tau$, it pushes the model toward harder questions, counteracting the growth of $\mathcal{J}_s(\theta)$. In this way, $R_q(z, q)$ balances the training dynamics, guiding the policy model toward generating questions of moderate difficulty.

Since questions with relatively large $p(q)$ can be estimated more accurately using the majority-voting–based estimation method, we set $\tau$ to a relatively large value to encourage the model to generate more questions $q$ with high $p(q)$. In our experiments, we set $\tau = \frac{3}{4}$.

**Question Filtering**   To ensure the quality of generated questions $q$, we further filter out some questions $q$ by setting $R_q(z, q) = 0$ and $R_s(q, s) = 0$. The filtering is based on the following rules:

1. Filter questions containing the word "prove".

2. Filter questions by multiple question marks, or paired keywords (e.g., "find . . . and . . ."), or enumerations (e.g., "1. . . . 2.").

3. Filter questions including "Solution:" or "Answer:" or "\boxed" or "The final answer is" or "To solve" or "Let's break down".

The first rule excludes mathematical proof questions, which are incompatible with majority voting in the solution reward function. The second rule removes questions containing multiple subquestions to ensure that each question has only a single answer. The third rule removes questions that include solutions, since we experimentally observe that their presence can shorten the model's output and potentially degrade performance.

**Mode Collapse**  Transitioning from semi-dynamic RL to dynamic RL introduces additional challenges, particularly the issues of mode collapse (Kossale et al., 2022). Mode collapse is a phenomenon in generative models in which the generator produces a limited variety of outputs, ignoring many modes of the true data distribution. In dynamic RL, mode collapse manifests in two forms: question mode collapse and answer mode collapse.

Question mode collapse occurs when the questions generated by the policy model are highly similar. For example, the model repeatedly generates questions like "Solve the equation $2x + 3 = 7$.". This lack of diversity can lead to saturation or even collapse in model performance. Therefore, it is essential to generate diverse questions that differ from one another.

Answer mode collapse arises when the answers extracted from model solutions are highly uniform. We observed that after a few hundred training steps, generated answers to different questions often converge to the same response, such as "\boxed{1}", causing performance collapse. This occurs because the golden answer is estimated via majority voting, which favors answers that are easily generated. Consequently, it is necessary to generate questions whose majority-voted golden answers are diverse.

**Diversity Reward Function**  To mitigate mode collapse, we propose diversity reward functions that encourage greater variability in generated data. The core idea is that the more an object resembles others, the lower its diversity reward. We first present the general form of the diversity reward function, followed by the specific formulations for question diversity reward function and answer diversity reward function.

Let $\{x_k\}_{k=1}^K$ be a set of $K$ objects sampled from the policy model $\pi_\theta$, where each object can be a question or an answer. For any pair of objects in this set, we define a similarity metric $S(\cdot, \cdot) \in [0, 1]$, where $S(x, x) = 1$ and larger values indicate greater similarity between objects. The diversity reward for an object $x \in \{x_k\}_{k=1}^K$ is then defined as

$$R_d(x) = \frac{1}{\sum_{k=1}^K S(x, x_k)}. \tag{9}$$

Intuitively, if many objects in the set are similar to $x$, the sum $\sum_{k=1}^K S(x, x_k)$ will be large, resulting in a smaller diversity reward $R_d(x)$. Conversely, a larger $R_d(x)$ indicates that $x$ is less similar to other objects. Since $0 \le S(x, x_k) \le 1$, it follows that $0 \le R_d(x) \le 1$, where $R_d(x) = 1$ implies that $x$ is only similar to itself.

The question diversity reward is then defined as

$$R_{dq}(q) = \frac{1}{\sum_{i=1}^n S_q(q, q_i)}, \tag{10}$$

where $q \in \{q_i\}_{i=1}^n$ and $S_q(\cdot, \cdot)$ measures the similarity between two questions. Each question $q$ is first tokenized into a sequence using a tokenizer $t(\cdot)$, and then similarity $S_q(\cdot, \cdot)$ is computed using the overlap ratio of sequences:

$$S_q(q, q_i) = \text{overlap}(t(q), t(q_i)) = \frac{\sum_{j=1}^{\min(|t(q)|, |t(q_i)|)} \mathbf{1}(t(q)_j = t(q_i)_j)}{\min(|t(q)|, |t(q_i)|)}$$

$$t(q) = [t(q)_1, t(q)_2, t(q)_3, \cdots], t(q_i) = [t(q_i)_1, t(q_i)_2, t(q_i)_3, \cdots]. \tag{11}$$

Similarly, the answer diversity reward is defined as

$$R_{da}(l(q)) = \frac{1}{\sum_{i=1}^n S_a(l(q), l(q_i))}, \tag{12}$$

where $l(\cdot)$ denotes the estimated golden answer defined in Eq. (4), and $S_a(\cdot, \cdot)$ is the similarity metric between two answers defined in Eq. (3).

**Objective Function**  Combining the solution reward function, question reward function, question diversity reward function, and answer diversity reward function, we define the following objective

---

**Algorithm 1** Dynamic RL: Dynamic Reinforcement Learning

---

**Input:** Initial policy model $\pi_{\theta_0}$, number of steps $N$, number of questions $n$, number of solutions $m$, prompt $z_0$, prompt $z_q$, coefficients $\{\lambda_q, \lambda_s, \lambda_{dq}, \lambda_{da}\}$, threshold $\tau$.
1: Initialize policy model $\pi_\theta \leftarrow \pi_{\theta_0}$.
2: **for** step $= 1$ **to** $N$ **do**
3:    Sample $n$ questions $q$ from $\pi_\theta(\cdot|z_0)$.
4:    Sample $m$ solutions $s$ from $\pi_\theta(\cdot|q, z_q)$ for each question $q$.
5:    Estimate the golden answer $l(q)$ by Eq. (4) for each question $q$.
6:    Compute solution reward $R_s(q, s)$ by Eq. (6) for each question-solution pair $(q, s)$.
7:    Compute question reward $R_q(z, q)$ Eq. (8) for each question $q$.
8:    Compute question diversity reward $R_{dq}(q)$ Eq. (10) for each question $q$.
9:    Compute answer diversity reward $R_{da}(l(q))$ Eq. (12) for each question $q$.
10:    Filter question by setting $R_q(z, q) = 0$ and $R_s(q, s) = 0$ according to the rules in Paragraph "Question Filtering" of Section 2.
11:    Optimize $\pi_\theta$ by maximizing objective function Eq. (14) via gradient ascent method.
12: **end for**
**Output:** Optimized policy model $\pi_\theta$.

---

function $\mathcal{L}(\theta)$:

$$\mathcal{L}(\theta) = \mathcal{L}_q(\theta) + \mathcal{L}_s(\theta),$$

$$\mathcal{L}_q(\theta) = \mathbb{E}_{z \sim p(z),\, q \sim \pi_\theta(\cdot|z)} \Big[ \lambda_q R_q(z, q) + \lambda_{dq} R_{dq}(q) + \lambda_{da} R_{da}(l(q)) \Big],$$

$$\mathcal{L}_s(\theta) = \mathbb{E}_{q \sim \pi_\theta(\cdot|z),\, s \sim \pi_\theta(\cdot|q)} \Big[ \lambda_s R_s(q, s) \Big], \tag{13}$$

where $\lambda_q, \lambda_{dq}, \lambda_{da}, \lambda_s \geq 0$ are coefficients to balance different reward functions.

By applying policy gradient theorem (Sutton et al., 1999) and reward normalization (Shao et al., 2024) to $\mathcal{L}_q(\theta)$ and $\mathcal{L}_s(\theta)$, we optimize the following surrogate objective function $\mathcal{J}(\theta)$,

$$\mathcal{J}(\theta) = \mathbb{E}_{z \sim p(z),\, q \sim \pi_\theta(\cdot|z)} \Big[ A_q \log \pi_\theta(q|z) \Big] + \mathbb{E}_{q \sim \pi_\theta(\cdot|z),\, s \sim \pi_\theta(\cdot|q)} \Big[ A_s \log \pi_\theta(s|q) \Big],$$

$$A_q = \lambda_q \frac{R_q(z, q) - \text{Mean}(\{R_q(z, q_i)\}_{i=1}^n)}{\text{Std}(\{R_q(z, q_i)\}_{i=1}^n)} + \lambda_{dq} \frac{R_{dq}(q) - \text{Mean}(\{R_{dq}(q_i)\}_{i=1}^n)}{\text{Std}(\{R_{dq}(q_i)\}_{i=1}^n)}$$

$$+ \lambda_{da} \frac{R_{da}(l(q)) - \text{Mean}(\{R_{da}(l(q_i))\}_{i=1}^n)}{\text{Std}(\{R_{da}(l(q_i))\}_{i=1}^n)},$$

$$A_s = \lambda_s \frac{R_s(q, s) - \text{Mean}(\{R_s(q, s_j)\}_{j=1}^m)}{\text{Std}(\{R_s(q, s_j)\}_{j=1}^m)}. \tag{14}$$

We use GRPO (Shao et al., 2024) to normalize each reward function and use advantage decomposition (Xiao et al., 2025) to get the final advantage function, which separately normalizes each individual reward function. Note that we treat $\pi_\theta(\cdot|z)$ as a fixed distribution when applying policy gradient theorem to $\mathcal{L}_s(\theta)$, analogous to semi-dynamic RL.

We present the detailed implementation of our dynamic RL in Alg.(1).

## 3 EXPERIMENTS

### 3.1 SETTINGS

**Baselines** We compare dynamic RL with semi-dynamic RL (Guo et al., 2025). For semi-dynamic RL, models are trained on four datasets of different scales: the small-scale MATH-7.5K (Hendrycks et al., 2021), the medium-scale DAPO-Math-17k (Yu et al., 2025), and the large-scale DeepScaleR-Preview-40K (Luo et al., 2025). In contrast, dynamic RL requires no external datasets, as it learns from its own generated data. We adopt Qwen2.5-Math-1.5B and Qwen2.5-Math-7B as the base models.

Table 1: The model performance on math datasets.

| Methods | MATH500 | AMC23 | AIME2024 | AIME2025 | Average |
|---|---|---|---|---|---|
| *Qwen2.5-Math-1.5B* | | | | | |
| Base Model | 40.8 | 24.2 | 4.4 | 4.2 | 18.4 |
| Semi-dynamic RL (7.5K) | 67.6 | 54.2 | 13.3 | 6.0 | 35.3 |
| Semi-dynamic RL (17K) | 71.2 | 50.3 | **17.9** | 7.5 | 36.7 |
| Semi-dynamic RL (40K) | 74.6 | **56.6** | 15.6 | 10.6 | **39.4** |
| Dynamic RL | **76.3** | 53.6 | 11.8 | **11.1** | 38.2 |
| *Qwen2.5-Math-7B* | | | | | |
| Base Model | 54.4 | 37.7 | 13.3 | 6.7 | 28.0 |
| Semi-dynamic RL (7.5K) | 76.2 | 60.8 | 24.8 | 11.0 | 43.2 |
| Semi-dynamic RL (17K) | 81.0 | 66.4 | 27.9 | 13.1 | 47.1 |
| Semi-dynamic RL (40K) | 81.2 | 64.5 | **29.2** | 16.3 | **47.8** |
| Dynamic RL | **83.8** | **66.9** | 21.4 | **17.3** | 47.4 |

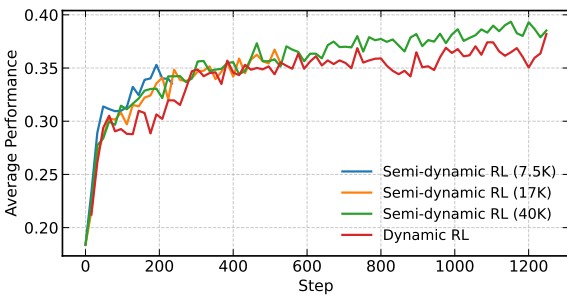

Figure 2: The average performance across different training steps.

**Evaluation**   We evaluate models on four math benchmark datasets: MATH500 (Hendrycks et al., 2021; Lightman et al., 2023), AMC23 (Art of Problem Solving, 2025b), AIME2024 and AIME2025 (Art of Problem Solving, 2025a). We report the avg@16 evaluation metric, which averages pass@1 over 16 sampled answers.

**Hyperparameters Settings**   We set $\lambda_s = 1$ and search $\lambda_q$ over $\{10^{-2}, 10^{-3}, 10^{-4}, 10^{-5}\}$. To promote answer diversity, we set $\lambda_{da} = 1$. For $\lambda_{dq}$, we conduct a search over $\{1, 10^{-1}, 10^{-2}, 10^{-3}\}$. For $\tau$ in Eq. (8), we search over $\{\frac{1}{4}, \frac{1}{2}, \frac{3}{4}\}$. The best-performing hyperparameters are found to be $\{\lambda_q = 10^{-3}, \lambda_{dq} = 10^{-1}, \tau = \frac{3}{4}\}$.

We set the batch size $n$ to 32, the number of sampled solutions $m$ to 16, and the learning rate to $10^{-6}$. For rollouts, we use temperature $= 1.0$ and top-$p = 1.0$, while for evaluation we use temperature $= 0.6$, top-$p = 0.95$ and top-$k = 20$. The maximum question length is set to 1024 tokens, and the maximum solution length is set to 3072 tokens. We train semi-dynamic for 1 epoch and dynamic RL the same steps as semi-dynamic trained on DeepScaleR-Preview-40K dataset.

To ensure fair comparison, we keep all shared hyperparameters identical between dynamic RL and semi-dynamic RL. The only differences lie in the methods themselves and the training datasets.

### 3.2   RESULTS

See Table 1 and Figure 2 for detailed results. Dynamic RL achieves performance comparable to semi-dynamic RL (40K) without relying on any external datasets, and it continues to improve over more than one thousand training steps. In contrast, Zhang et al. (2025c) reports that semi-dynamic RL without golden answers can sustain improvement for only a limited number of steps.

Table 2: The model performance on math datasets.

| Methods | MATH500 | AMC23 | AIME2024 | AIME2025 | Average |
|---|---|---|---|---|---|
| Dynamic RL (40K) | 83.8 | 66.9 | 21.4 | 17.3 | 47.4 |
| w/o filtering | 81.6 | 66.7 | 20.8 | 15.6 | 46.2 |
| w/o $R_q$ | 82.2 | 62.3 | 19.2 | 16.3 | 45.0 |
| w/o $R_{dq}$ | 82.4 | 64.5 | 14.6 | 14.0 | 43.9 |
| w/o $R_{da}$ | 80.4 | 66.4 | 19.0 | 13.5 | 44.8 |
| $\tau = 1/4$ | 82.8 | 65.5 | 20.6 | 18.8 | 46.9 |
| $\tau = 3/4$ | 81.2 | 64.7 | 20.2 | 19.6 | 46.4 |
| $\lambda_{dq} = 1$ | 82.2 | 63.3 | 19.5 | 17.5 | 45.7 |
| $\lambda_{dq} = 10^{-5}$ | 83.2 | 65.3 | 16.9 | 16.0 | 45.4 |
| $\lambda_q = 1$ | 81.2 | 63.8 | 20.4 | 14.2 | 44.9 |
| $\lambda_q = 10^{-5}$ | 81.8 | 62.3 | 16.0 | 14.6 | 43.7 |

## 3.3 ABLATION STUDIES

We conduct ablation studies to examine the effectiveness of the question filtering, the question reward function $R_q$, the question diversity reward function $R_{dq}$, the answer diversity reward function $R_{da}$, and the choice of $\tau$ in Eq. (8).

As reported in Table 2, all three reward functions and question filtering contribute to improving model performance. Moreover, setting a relatively large value of $\tau$ facilitates more reliable estimation of answers, which in turn leads to better performance.

## 3.4 EXPLORATION AND EXPLOITATION

We further demonstrate the effectiveness of dynamic RL from the view of exploration and exploitation. The objective function in Eq. (14) consists of four reward functions. We temporarily omit the answer diversity reward function $R_{da}$ because answer diversity is relatively easier to satisfy. The solution reward function $R_s$ plays a role for improving the model performance, similar to semi-dynamic RL. The question diversity reward function $R_{dq}$ plays a role of exploration, which aims to generate new questions. The question reward function $R_q$ plays a role of exploitation, which favors questions with proper $p(q)$ (See Figure 1).

To balance exploration and exploitation, we tune the hyper-parameters $\lambda_{dq}$ and $\lambda_q$. For $\lambda_{dq}$, an excessively large value causes the model to prioritize generating new questions, preventing convergence, while too small a value may lead to mode collapse in question generation and, consequently, degraded model performance. Table 2 illustrates the performance across different values of $\lambda_{dq}$. In practice, we set $\lambda_{dq}$ to a relatively large value to preserve sufficient exploration.

For $\lambda_q$, we adopt a relatively small value to moderate exploitation. If $\lambda_q$ is too small, $R_{dq}$ dominates, resulting in over-exploration. Conversely, if $\lambda_q$ is too large, the model generates overly hard questions, making it impossible for the LLM to accurately estimate golden answers, which ultimately harms performance as shown in Table 2.

To further clarify this point, Figure 3 presents the distribution of $r(q)$ (as defined in Eq.(5)) at different training steps, where $r(q)$ denotes the frequency of the majority answer. We focus on the red curve corresponding to $r(q) = m = 16$, as this category constitutes the largest portion of each batch and reflects the simplest type of questions. The number of questions satisfying $r(q) = m$ therefore serves as a proxy for batch-level difficulty: the higher this number, the easier the questions in the batch.

As shown in Figure 3, larger values of $\lambda_q$ reduce the count of questions with $r(q) = m$, because $R_q(z, q) = 0$ when $r(q) = m$. As a result, an excessively large $\lambda_q$ pushes the question generator too quickly toward harder questions, making the answer estimation increasingly unreliable. In summary, the difficulty of the generated questions at each training step should remain aligned with the current capability of the LLM and the reliability of the chosen estimation method.

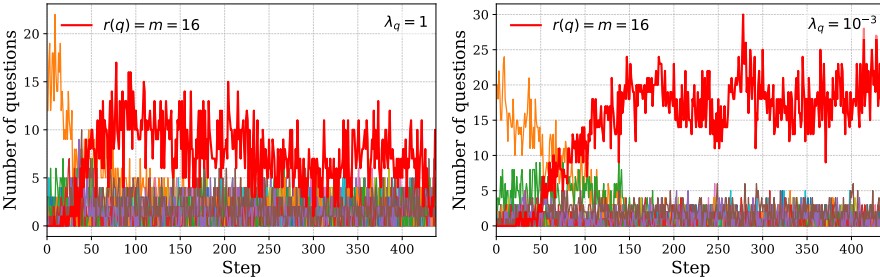

Figure 3: The distribution of $r(q)$ during training. The curves show the counts of questions with $r(q) = i$ for $1 \leq i \leq 16$, reflecting the full distribution of $r(q)$ throughout training. Since it is difficult to derive meaningful conclusions from curves other than the one for $r(q) = m = 16$, we have intentionally omitted legends for the remaining lines.

We allow the model to generate a proportion non-contributory questions ($r(q) = m$, the advantage $A_s = 0$ (defined in Eq.(14))), such that the remaining questions are located near the boundary $r(q) = m$. Learning from data near the boundary may extend the boundary of an LLM by itself.

## 4 RELATED WORK

**Unsupervised RL** DeepSeek-R1 (Guo et al., 2025) shows that RL can significantly enhance model performance without relying on human-labeled solutions. Nevertheless, it still depends on human-labeled golden answers to guide the learning process. In contrast, unsupervised RL seeks to train models entirely without human-labeled answers. Some approaches define objectives based on the consistency of model outputs, Zhang et al. (2025a) propose rewards derived from intermediate reasoning states, and Zuo et al. (2025) explore estimating golden answers through answer consistency. Meanwhile, Shao et al. (2025) demonstrate that even random or negative rewards can serve as effective training signals for RL. Other approaches, including Zhang et al. (2025b) and Agarwal et al. (2025), link RL with entropy minimization at either the sequence or token level, using answer entropy as a surrogate objective. These methods primarily focus on estimation strategies. In contrast, dynamic RL emphasizes generating questions that can be reliably assessed by estimation methods.

**Self-play RL** Self-play RL is a paradigm in which an agent enhances its performance by iteratively interacting with versions of itself (Zhang et al., 2024). This approach typically relies on a verifiable environment, particularly in code-related tasks, where unit tests can provide efficient verification. Lin et al. (2025) propose a self-play solver-verifier framework that jointly improves a model's ability to generate both code and corresponding test units. Similarly, Wang et al. (2025) introduce a framework that co-evolves coding and unit test generation by leveraging feedback from their interactions. Other studies, such as Zhao et al. (2025) and Zhou et al. (2025), allow the LLM to generate code tasks and learn from them, provided these tasks are feasible and verifiable. Despite these advances, these methods still depend on an external verifier to check answer correctness, whereas dynamic RL estimates the golden answers directly from the model's outputs.

## 5 CONCLUSION

In this paper, we propose dynamic RL, a framework designed to enhance the scalability of RL by leveraging the LLM itself. Dynamic RL samples both questions and solutions directly from the LLM, allowing the training data to adapt dynamically as the model evolves. However, this approach introduces new challenges, including the absence of golden answers and the risk of mode collapse. To address these issues, we encourage the model to generate a greater proportion of questions that can be reliably estimated and introduce a diversity reward function to promote data diversity. Experimental results demonstrate that Dynamic RL achieves performance comparable to semi-dynamic RL, without relying on external supervision. We envision that further refinements of this approach will continue to improve the scalability of RL.

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

# A APPENDIX

## A.1 QUESTIONS

Table 3 presents several generated questions whose answers may not be accurately estimated.

Table 3: Examples of issues in generated questions.

| Issue | Case | Analysis |
|---|---|---|
| Non-unique Answer (No Answer) | Find two integers $x$ and $y$ such that $x^2 + y^2 = 7$. | This question has no valid answer. The model may still output an arbitrary answer, leading to errors. Since there is no general method to determine whether a mathematical problem has a unique solution, we add the prompt "The answer to the question should exist and be unique." to the prompt $z_0$ to reduce the occurrence of such cases. |
| Non-unique Answer (Multiple Answers) | Find the unique positive integer $x$ such that $\lfloor \frac{x}{5} \rfloor \times \lfloor \frac{x}{7} \rfloor = 15$. | The correct answers are $\{25, 26, 27\}$, so the answers are not unique. However, the model may output only one answer (e.g., 25) and treat it as the golden answer. Consequently, other correct answers may be incorrectly judged as wrong. |
| Non-unique Answer (Insufficient Conditions) | What is the determinant of a 2x2 matrix? | This question has insufficient conditions because the matrix itself is not given. The model may compute determinants for different matrices, producing inconsistent answers. Such questions tend to be treated as hard because $p(q)$ may decrease when multiple answers exist. Therefore, we require that every question admit a unique answer. |
| Sub-questions | In triangle ABC, vertex A has an angle of $163°$, side BC measures 1.9 units, and vertex C has an angle of $7°$. Find: 1. The area of triangle ABC. 2. The circumradius of triangle ABC. 3. The inradius of triangle ABC. 4. The semiperimeter of triangle ABC. | This question contains multiple sub-questions. The model may output only one result or place only one in "\boxed", which lowers $p(q)$. Consequently, such questions are often treated as hard due to low $p(q)$. We therefore apply a filtering rule to remove this type of questions. |

| Issue | Case | Analysis |
|---|---|---|
| Numerical Approximation | Suppose the probability that a randomly chosen 10-digit phone number contains at least one digit that is 1 is calculated. If there are 9 choices for digits 2-9 and 10 choices for digit 0, find this probability. Also, consider if each digit is used independently for the phone number. | The correct answer is $1 - \left(\frac{9}{10}\right)^{10} \approx 0.6513$. Answers such as $0.6513$ and $0.651$ are mathematically correct, yet minor numerical differences may cause them to be treated as incorrect. Since such cases occur infrequently, we do not explicitly address them. |
| Underestimated Difficulty | Let $X$ and $Y$ be independent random variables, each following a uniform distribution on the interval $[0,1]$. Let $Z = \min(X,Y)$ and $W = \max(X,Y)$. Find the value of $P(Z + W \geq 1.5)$. | We first demonstrate that the metric $p(q)$ only approximates question difficulty and may underestimate it in some cases. Let $l_t(q)$ denote the true golden answer, $r_t(q)$ the number of answers equivalent to it, and $p_t(q) = \frac{r_t(q)}{m}$ the true difficulty measure. Since $r(q)$ in Eq. (5) is maximized, $p(q)$ serves as an upper bound of $p_t(q)$, i.e., $p_t(q) \leq p(q)$. For this question, the correct answer is $\frac{1}{8}$. However, the frequency of "$\frac{1}{8}$" is 3, while that of "$\frac{1}{2}$" is 4, yielding $p(q) = \frac{4}{16}$, which exceeds the true measure $p_t(q) = \frac{3}{16}$. Thus, using $p(q)$ can underestimate difficulty and mislead majority voting. To mitigate this, we set $\tau$ in Eq. (8) to a relatively high value and tune hyperparameters for more reliable question generation. |

Table 4 presents several questions generated at different training steps that can be correctly answered.

Table 4: Questions across training steps.

| Step | Question | Solution |
|------|----------|----------|
| 1 | What is the value of x in the equation $2x + 3 = 7$? | To find the value of $x$ in the equation $2x + 3 = 7$, we need to isolate $x$. Here are the steps: 1. Subtract 3 from both sides of the equation: $$2x + 3 - 3 = 7 - 3$$ Simplifying, we get: $$2x = 4$$ 2. Divide both sides of the equation by 2: $$\frac{2x}{2} = \frac{4}{2}$$ Simplifying, we get: $$x = 2$$ So, the value of $x$ is $\boxed{2}$. |

| Step | Question | Solution |
|------|----------|----------|
| 50 | Solve for $x$: $$x^2 - 5x + 6 = 0$$ | To solve the quadratic equation $x^2 - 5x + 6 = 0$, we can use the quadratic formula, which is given by: $$x = \frac{-b \pm \sqrt{b^2 - 4ac}}{2a}$$ where $a = 1$, $b = -5$, and $c = 6$. Alternatively, we can factorize the quadratic equation directly. Let's try both methods. ### Method 1: Factorization We need to find two numbers that multiply to give 6 (the constant term) and add up to -5 (the coefficient of the $x$ term). The numbers -2 and -3 meet these conditions because $(-2) \times (-3) = 6$ and $(-2) + (-3) = -5$. So we can write: $$x^2 - 5x + 6 = (x-2)(x-3)$$ Setting each factor to zero gives us: $$x - 2 = 0 \quad \text{or} \quad x - 3 = 0$$ So the solutions are: $$x = 2 \quad \text{or} \quad x = 3$$ ### Method 2: Quadratic Formula Using the quadratic formula: $x = \frac{-(-5) \pm \sqrt{(-5)^2 - 4 \cdot 1 \cdot 6}}{2 \cdot 1} = \frac{5 \pm \sqrt{25-24}}{2} = \frac{5 \pm \sqrt{1}}{2} = \frac{5 \pm 1}{2}$ This gives us two solutions: $x = \frac{5+1}{2} = \frac{6}{2} = 3$ $x = \frac{5-1}{2} = \frac{4}{2} = 2$ So the solutions are: $$x = 2 \quad \text{or} \quad x = 3$$ The final answer is: $$\boxed{2, 3}$$ |

| Step | Question | Solution |
|------|----------|----------|
| 100 | Consider three sets $A$, $B$, and $C$ defined as follows: $$A = \{x \in \mathbb{R} \mid x^2 - 3x + 2 \geq 0\}$$ $$B = \{x \in \mathbb{R} \mid x < 2\}$$ $$C = \{x \in \mathbb{R} \mid x^2 - 4 \leq 0\}$$ Determine $(A \cup B) \cap C$. | To solve the problem, let's first determine the elements of each set $A$, $B$, and $C$. 1. **Set $A$**: The inequality $x^2 - 3x + 2 \geq 0$ can be factored as $(x-1)(x-2) \geq 0$. This inequality holds when $x \leq 1$ or $x \geq 2$. So, $A = (-\infty, 1] \cup [2, \infty)$. 2. **Set $B$**: This is the set of all real numbers less than 2, i.e., $B = (-\infty, 2)$. 3. **Set $C$**: The inequality $x^2 - 4 \leq 0$ can be factored as $(x-2)(x+2) \leq 0$. This inequality holds when $-2 \leq x \leq 2$. So, $C = [-2, 2]$. Next, we need to find $A \cup B$: $A \cup B = (-\infty, 1] \cup [2, \infty) \cup (-\infty, 2) = (-\infty, 2) \cup [2, \infty) = (-\infty, \infty) = \mathbb{R}$. Finally, we need to find $(A \cup B) \cap C$: $(A \cup B) \cap C = \mathbb{R} \cap [-2, 2] = [-2, 2]$. So the final answer is $\boxed{[-2, 2]}$. |

| Step | Question | Solution |
|---|---|---|
| 500 | Evaluate the line integral of the vector field $$\vec{F}(x,y) = (y^2, x^2)$$ along the curve defined by the parametric equations $x = \cos(t), \quad y = \sin(t), \quad 0 \le t \le 2\pi$. | To evaluate the line integral of the vector field $\vec{F}(x,y) = (y^2, x^2)$ along the curve defined by the parametric equations $x = \cos(t)$, $y = \sin(t)$ for $0 \le t \le 2\pi$, we follow these steps: 1. **Parameterize the curve**: The curve is given by $x = \cos(t)$ and $y = \sin(t)$. The vector function for the curve is $\vec{r}(t) = (\cos(t), \sin(t))$. 2. **Find the derivative of the parameterization**: The derivative of $\vec{r}(t)$ with respect to $t$ is $\vec{r}'(t) = (-\sin(t), \cos(t))$. 3. **Evaluate the vector field along the curve**: Substitute $x = \cos(t)$ and $y = \sin(t)$ into the vector field $\vec{F}(x,y) = (y^2, x^2)$. This gives $\vec{F}(\cos(t), \sin(t)) = (\sin^2(t), \cos^2(t))$. 4. **Compute the dot product**: The dot product $\vec{F}(\cos(t), \sin(t)) \cdot \vec{r}'(t)$ is $(\sin^2(t), \cos^2(t)) \cdot (-\sin(t), \cos(t)) = -\sin^3(t) + \cos^3(t)$. 5. **Set up the integral**: The line integral is $\int_0^{2\pi} (-\sin^3(t) + \cos^3(t))\, dt$. 6. **Evaluate the integral**: We can split the integral into two parts: $\int_0^{2\pi} -\sin^3(t)\, dt + \int_0^{2\pi} \cos^3(t)\, dt$. Both $\sin^3(t)$ and $\cos^3(t)$ are odd functions over the interval $[0, 2\pi]$, and the integral of an odd function over a symmetric interval around the origin (or any multiple of the period) is zero. Therefore, both integrals are zero: $\int_0^{2\pi} -\sin^3(t)\, dt = 0$ and $\int_0^{2\pi} \cos^3(t)\, dt = 0$. Adding these together, we get $\int_0^{2\pi} (-\sin^3(t) + \cos^3(t))\, dt = 0$. The final answer is $\boxed{0}$. |

| Step | Question | Solution |
|------|----------|----------|
| 1000 | Given the function $f(v) = v^5 + v^3$, find the integral of $\frac{f'(v)}{2f(v)}$ with respect to $v$. | To find the integral of $\frac{f'(v)}{2f(v)}$ with respect to $v$, we start by identifying the function $f(v)$ and its derivative $f'(v)$. Given $f(v) = v^5 + v^3$, we compute the derivative: $f'(v) = \frac{d}{dv}(v^5 + v^3) = 5v^4 + 3v^2$. Now, we substitute $f'(v)$ and $f(v)$ into the integral: $\int \frac{f'(v)}{2f(v)}\, dv = \int \frac{5v^4+3v^2}{2(v^5+v^3)}\, dv$. We can simplify the integrand by factoring out the common terms in the numerator and the denominator: $\int \frac{5v^4+3v^2}{2(v^5+v^3)}\, dv = \int \frac{5v^4+3v^2}{2v^3(v^2+1)}\, dv = \int \frac{5v+3/v}{2(v^2+1)}\, dv$. This integral can be simplified further by recognizing that the numerator $5v + 3/v$ can be split into two separate fractions: $\int \frac{5v+3/v}{2(v^2+1)}\, dv = \int \left( \frac{5v}{2(v^2+1)} + \frac{3/v}{2(v^2+1)} \right)\, dv = \int \frac{5v}{2(v^2+1)}\, dv + \int \frac{3/v}{2(v^2+1)}\, dv$. However, a simpler approach is to recognize that the integral $\int \frac{f'(v)}{2f(v)}\, dv$ is of the form $\int \frac{1}{2}\frac{f'(v)}{f(v)}\, dv$, which is a standard form that can be integrated directly. The integral of $\frac{f'(v)}{f(v)}$ is $\ln|f(v)|$, so we have: $\int \frac{f'(v)}{2f(v)}\, dv = \frac{1}{2}\int \frac{f'(v)}{f(v)}\, dv = \frac{1}{2}\ln|f(v)| + C$, where $C$ is the constant of integration. Substituting back the function $f(v) = v^5 + v^3$, we get: $\int \frac{f'(v)}{2f(v)}\, dv = \frac{1}{2}\ln|v^5+v^3| + C$. Therefore, the final answer is: $$\boxed{\frac{1}{2}\ln|v^5+v^3| + C}$$ |

## A.2 EXPERIMENTS

We further show more experimental results.

**Training Dynamics** We show the training dynamics of different rewards as in Figure 4.

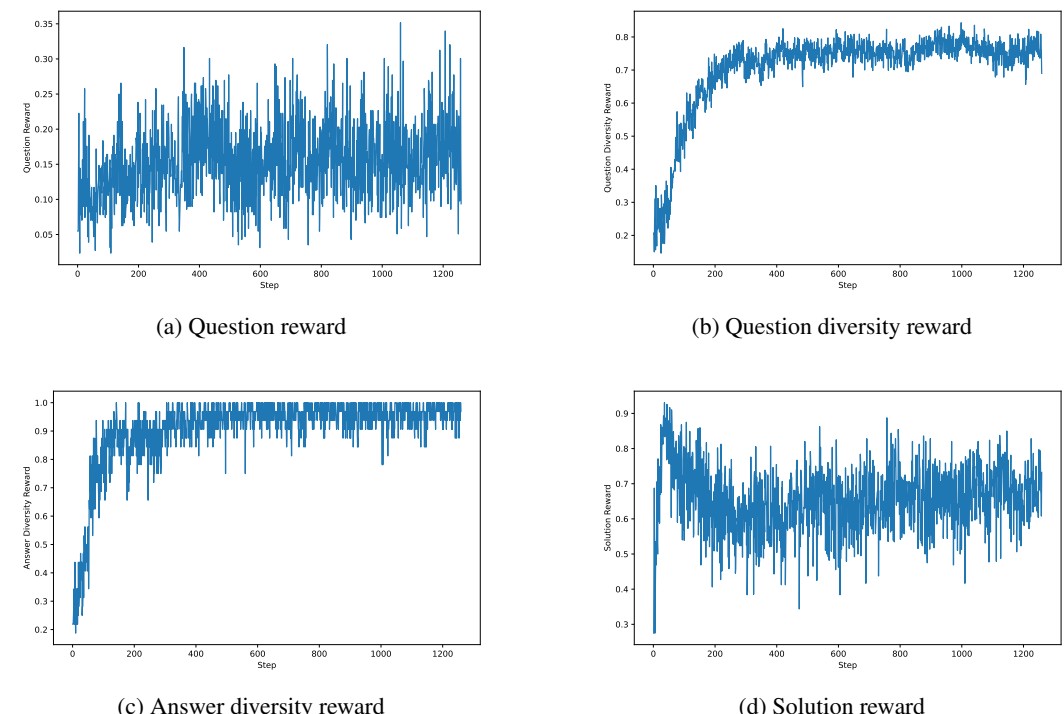

Figure 4: Overview of four reward components.

The answer diversity reward steadily increases and eventually remains close to 1, as satisfying answer diversity is relatively easy. The question diversity reward also increases during the early stages of training and then stabilizes in the range of approximately 0.7 to 0.8, which prevents both mode collapse and excessive exploration.

The trajectories of the question reward and solution reward highlight the adversarial relationship between these two objectives. In the early phase of training (Step 0 to Step 50), the model performance improves rapidly, leading to a sharp increase in the solution reward, while the question reward decreases because overly simple questions receive low scores. As training progresses (Step 50 to Step 450), the generated questions gradually become more challenging, causing the solution reward to decline and the question reward to rise. After this phase, both rewards continue to fluctuate within a stable range, maintaining an effective balance between question difficulty and model solvability.

**Mode Collapse** We further empirically analyze the mode collapse phenomena. Figure 5 illustrates the entropy of generated questions with and without the question diversity reward. Without this reward, the entropy becomes unstable, collapsing around Step 750 and then exploding around Step 870. Since entropy alone may not fully capture mode collapse, we also manually inspected the generated questions. We found that 93.75% of the questions became equivalent by Step 750, and by Step 870 the model produced largely random tokens. In contrast, when the question diversity reward is enabled, we do not observe the mode collapse phenomenon.

Figure 6 presents the ratio of unique answers within each batch, with and without the answer diversity reward. Without this reward, the ratio collapses to nearly zero after Step 480. With the reward enabled, the ratio increases to nearly one by Step 200. Manual inspection further confirms degeneration without this reward: by Step 480, 96.9% of answers collapse to the answer "0".

**Sensitivity Analysis** We conduct sensitivity analysis on the hyperparameter $\lambda_{dq}$ and $\lambda_q$. See Table 5 and Table 6 for the results. It can be seen that the performance does not oscillate with changes in $\lambda_q$ and $\lambda_{dq}$.

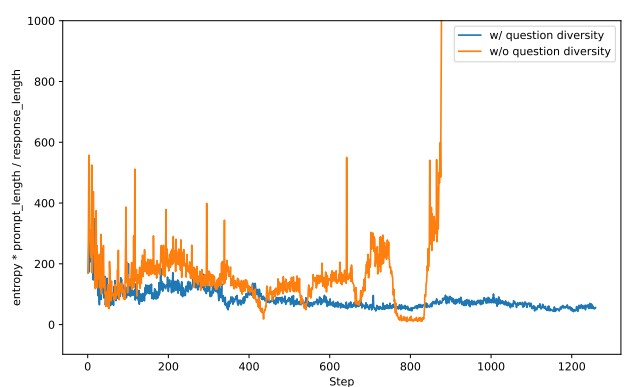

Figure 5: The entropy of generated questions during training.

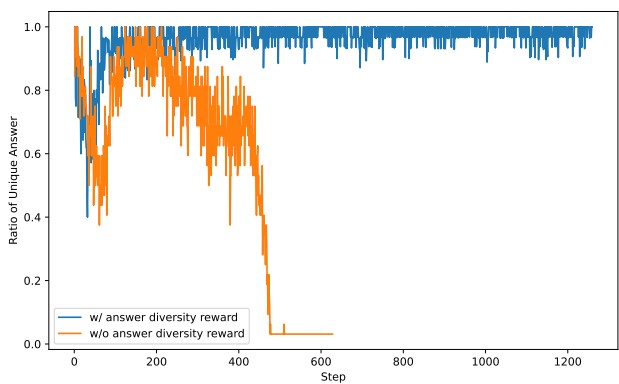

Figure 6: The ratio of unique answer during training.

Table 5: The model performance with different $\lambda_q$.

| $\lambda_q$ | 1 | $10^{-1}$ | $10^{-2}$ | $10^{-3}$ | $10^{-4}$ | $10^{-5}$ |
|---|---|---|---|---|---|---|
| **MATH500** | 81.2 | 82.6 | 82.4 | 83.8 | 82.0 | 81.8 |
| **AMC23** | 63.8 | 65.3 | 68.9 | 66.9 | 66.1 | 62.3 |
| **AIME24** | 20.4 | 19.2 | 17.1 | 21.4 | 19.8 | 16.0 |
| **AIME25** | 14.2 | 16.5 | 16.9 | 17.3 | 14.8 | 14.6 |
| **Average** | 44.9 | 45.9 | 46.3 | 47.4 | 45.7 | 43.7 |

Table 6: The model performance with different $\lambda_{dq}$.

| $\lambda_q$ | 1 | $10^{-1}$ | $10^{-2}$ | $10^{-3}$ | $10^{-4}$ | $10^{-5}$ |
|---|---|---|---|---|---|---|
| **MATH500** | 82.2 | 83.8 | 83.0 | 81.4 | 81.4 | 83.2 |
| **AMC23** | 63.3 | 66.9 | 65.0 | 66.3 | 65.3 | 65.3 |
| **AIME24** | 19.5 | 21.4 | 19.2 | 19.4 | 20.8 | 16.9 |
| **AIME25** | 17.5 | 17.3 | 18.5 | 16.0 | 16.6 | 16.0 |
| **Average** | 45.7 | 47.4 | 46.4 | 45.8 | 45.5 | 45.4 |

