# OpenReview forum: "All by Large Language Model Itself"
_ICLR.cc/2026/Conference — ICLR 2026 Conference Withdrawn Submission_

### Official Review · Reviewer_ZtoL · 2025-11-01

**Soundness:** 2
**Presentation:** 3
**Contribution:** 2
**Rating:** 4
**Confidence:** 4

**Summary:**

This paper proposes dynamic reinforcement learning (dynamic RL), a self-contained training framework for large language models (LLMs) in which both questions and solutions are generated and refined by the model itself, without reliance on external datasets or human-labeled answers. The method builds upon semi-dynamic RL (e.g., DeepSeek-R1), but eliminates the need for static human-authored questions. To address the absence of ground-truth answers, the authors estimate “golden answers” via majority voting over multiple sampled solutions per question. The paper introduce four reward components:Solution reward, Question reward, Question diversity reward and Answer diversity reward. Results show that dynamic RL achieves performance comparable to semi-dynamic RL trained on large-scale human-curated datasets, despite using only self-generated data.

**Strengths:**

- The paper is well-structured and easy to follow.  The framework is described with mathematical rigor and algorithmic transparency.
- This paper proposes a fully autonomous RL loop for LLMs, removing dependence on any external data—a bold step toward “self-improving” systems.

**Weaknesses:**

1. **Unverified estimation quality**: No analysis of how often majority voting recovers the *true* answer. Table 3 shows cases where it fails, but no aggregate statistics (e.g., % of questions with correct $l(q)$) are given.

2. **Inadequate validation of diversity mechanisms**: The paper claims to mitigate mode collapse but provides no direct evidence. There is no comparison of question/answer entropy with vs. without $R_{dq}, R_{da}$, or visualization of question embedding clusters over time, or measurement of repetition rates (e.g., % of duplicate or near-duplicate questions).

3. **Flawed question similarity metric**: Token overlap (Eq. 11) is **not semantically meaningful**. This undermines the validity of $R_{dq}$. e.g.: Consider two questions: "Solve for \(x\): \(2x + 3 = 7\)." and "Find the value of \(x\) that satisfies \(2x + 3 = 7\)." These are semantically identical, but their token sequences differ significantly. The overlap-based similarity in Eq. (11) would deem them dissimilar, causing the diversity reward \(R_{dq}\) to incorrectly encourage redundancy. Conversely, questions with high token overlap but different meanings (e.g., same variable name but different equations) may be wrongly penalized. Thus, token overlap is not a reliable measure of semantic similarity. A more robust metric (e.g., embedding similarity) would strengthen claims.

4. **Incomplete ablation study**: The ablation (Table 2) removes $R_q$, $R_{dq}$, $R_{da}$, but don't remove $R_s$. It needs to be tested whether learning is truly driven by self-consistency or just diversity/exploitation heuristics.

5. **Lack of training dynamics analysis**: The paper does not show how the four reward signals co-evolve during training, for example, whether they are balanced or a specific one dominates. This is essential to understand the optimization landscape.

**Questions:**

1. **Estimation accuracy**: What is the empirical accuracy of the majority-voted golden answer $l(q)$ against ground truth (e.g., on a subset of generated questions manually verified)?

2. **Question reward efficacy**: Does the average $p(q)$ increase over training steps? If not, how do you reconcile this with the claim that $R_q$ encourages “estimable” questions?

3. **Mode collapse evidence**: Can you provide quantitative metrics of diversity (e.g., number of unique answers, question embedding variance) with and without $R_{dq}$ and $R_{da}$? Without this, the mitigation claim remains unsubstantiated.

4. **Ablation of $R_s$**: Why was the solution reward $R_s$ not ablated? Is learning possible without it?

---

> ### Author Response · Authors · 2025-11-20
>
> We appreciate your careful and constructive comments. We have addressed the questions that you raised as follows. Please let us know if you have any further concerns.
>
> $\textbf{Q1:}$ No analysis of how often majority voting recovers the true answer.
>
> $\textbf{A1:}$ We do not report the ratio of recovered golden answers for two reasons. First, verifying golden answers requires manual checking, which is not scalable. Second, a high recovery ratio does not necessarily indicate better question quality, because question difficulty strongly influences this ratio: golden answers for easier questions are more likely to be correctly estimated. However, continually training on overly simple questions can lead to performance degradation in practice.
>
> To obtain a practical proxy for question difficulty (which is related to the recovery ratio), we use the proportion of questions in a batch that satisfy $p(q)=1$ or $r(q)=m$ (i.e., all sampled answers for a question are identical) as an indicator of batch difficulty. This choice is motivated by empirical observation: we manually checked approximately 100 generated questions meeting \$p(q)=1$ or $r(q)=m$ and find that their estimated answers are all correct. Therefore, this indicator provides a relatively reliable measure of question difficulty.
>
> We then use this indicator to guide training. As shown in Figure 3, we adjust $\lambda_{q}$ to maintain a relatively high number of questions with $r(q)=m$. This ensures that the answers to a larger fraction of questions can be reliably estimated, and training on these questions improves the model’s performance, as demonstrated in Table 1.
>
> $\textbf{Q2:}$ Inadequate validation of diversity mechanisms.
>
> $\textbf{A2:}$ Due to space constraints, we place the validation of the diversity mechanisms in the appendix of the revision. Figure 5 (Page 20) illustrates the entropy of generated questions with and without the question diversity reward. Without this reward, the entropy becomes unstable, collapsing around Step 750 and then exploding around Step 870. Since entropy alone may not fully capture mode collapse, we also manually inspected the generated questions. We found that 93.75\% of the questions became equivalent by Step 750, and by Step 870 the model produced largely random tokens. In contrast, when the question diversity reward is enabled, we do not observe the mode collapse phenomena.
>
> Figure 6 (Page 20) presents the ratio of unique answers within each batch, with and without the answer diversity reward. Without this reward, the ratio collapses to nearly zero after Step 480. With the reward enabled, the ratio increases to nearly one by Step 200. Manual inspection further confirms degeneration without this reward: by Step 480, 96.9\% of answers collapse to the answer “0”.
>
> $\textbf{Q3:}$ Flawed question similarity metric. Token overlap (Eq. 11) is not semantically meaningful.
>
> $\textbf{A3:}$ In practice, the token overlap similarity metric does not suffer from issues with semantically equivalent questions. We were aware of this potential concern when selecting the overlap metric, but manual inspection of the generated questions did not find cases where semantically equivalent questions had a very low token overlap ratio.
>
> This observation can be explained from two perspectives. First, RL decreases the probability of data with relatively lower rewards. Semantically equivalent questions tend to have higher token overlap ratios compared to semantically inequivalent questions, which typically leads to lower rewards. Second, the token overlap metric aligns well with the autoregressive nature of LLMs, which encourages a relatively high generation entropy. When the policy model has high entropy or diversity, the set of questions that can be sampled becomes large. The subset of semantically equivalent questions with low token overlap is extremely small relative to this large set, making their probability of being sampled negligible.
>
> Since the question space is effectively infinite, allowing questions with high token overlap but different meanings (i.e., false positives) to be excluded does not limit the overall diversity.
>
> We select the token overlap metric because it is simple, efficient, and transparent in terms of when it may fail. While other metrics could potentially capture semantic equivalence in certain cases, they are more difficult to analyze and interpret when failures occur.

---

> ### Author Response · Authors · 2025-11-20
>
> $\textbf{Q4:}$ Incomplete ablation study for the solution reward.
>
> $\textbf{A4:}$ Since model performance is evaluated based on the quality of the solutions or answers generated for a given set of questions, the solution reward function is essential for improving solution quality under this evaluation framework. The average performance without the solution reward is 0.406, whereas incorporating the solution reward increases it to 0.474.
>
> In contrast, evaluating the quality of generated questions is inherently open-ended and depends heavily on the specific metric chosen (e.g., diversity or difficulty). For this reason, we do not provide a concrete quantitative evaluation of the questions.
>
> $\textbf{Q5:}$ Lack of training dynamics analysis.
>
> $\textbf{A5:}$ Due to space constraints, the training dynamics are included in the appendix of the revised version (Pages 19).
>
> The answer diversity reward steadily increases and eventually remains close to 1, as satisfying answer diversity is relatively easy. The question diversity reward also increases during the early stages of training and then stabilizes in the range of approximately 0.7 to 0.8, which prevents both mode collapse and excessive exploration.
>
> The trajectories of the question reward and solution reward highlight the adversarial relationship between these two objectives. In the early phase of training (Step 0 to Step 50), the model performance improves rapidly, leading to a sharp increase in the solution reward, while the question reward decreases because overly simple questions receive low scores. As training progresses (Step 50 to Step 450), the generated questions gradually become more challenging, causing the solution reward to decline and the question reward to rise. After this phase, both rewards continue to fluctuate within a stable range, maintaining an effective balance between question difficulty and model solvability.
>
> $\textbf{Q6:}$ Does the average $p(q)$ increase over training steps?
>
> $\textbf{A6:}$ We have already plotted the distribution of $r(q)$ during training in Figure 3 (where $r(q) = p(q)/m$ and $m = 16$). The vertical axis shows the number of questions in a batch that satisfy $(q) = i$ for $0 \leq i \leq m $. Since the maximum value of $p(q)$ is 1, it is sufficient to focus on the red curve, which corresponds to questions with $r(q) = m$ or equivalently $p(q) = 1$.
>
> We adjust the hyperparameter $\lambda_{q}$ to keep the number of questions with $p(q)=1$ relatively high (right panel of Figure 3). As shown in the figure, this number increases during the early stages of training and then stabilizes within a relatively large range.

---

### Official Review · Reviewer_HgKX · 2025-11-01

**Soundness:** 2
**Presentation:** 3
**Contribution:** 2
**Rating:** 4
**Confidence:** 4

**Summary:**

This paper proposes RL, in which the LLM autonomously generates and solves its own problems. Compared with the traditional RL framework, the proposed dynamic RL also provides a learning signal for the LLM in its role as a problem setter, guiding it to produce questions of moderate difficulty. Meanwhile, the LLM’s problem-solving ability continues to improve throughout training as well.

**Strengths:**

- The proposed method is clearly present and easy to follow.
- The topic of enabling LLMs to automatically generate high-quality question-answer pairs is important, especially as model capabilities continue to grow while high-quality, valuable question-answer pairs become increasingly scarce.

**Weaknesses:**

- The proposed method essentially leverages the LLM’s own **self-consistency** to provide learning signals for both its solver and question-setter roles. Unfortunately, such methods primarily reinforce the determinism of the model’s outputs rather than genuinely enhancing its reasoning ability, especially when it comes to solving challenging reasoning problems. For example, when the base model already possesses a reasonable level of reasoning capability, substantial further improvement typically requires RL or SFT training on competition-level problems (e.g., AIME). However, for such difficult problems, the LLM’s responses are mostly incorrect, and even majority voting is likely to yield wrong answers. Consequently, the proposed method cannot provide reliable learning signals for these cases, nor can it effectively encourage the generation of high-quality, high-difficulty problems.
- The experimental results partly confirm my above opinion: on AIME24 and AIME25, the proposed method achieves only limited gains. Under a well-designed prompt template, Qwen2.5-Math-1.5B base can already reach a score of 16.7 on AIME 2024 [1], which is higher than that achieved by Dynamic RL. On simpler benchmarks, Dynamic RL performs better, likely because it strengthens confidence in answers that are already mostly correct.
- The method resembles multi-objective RL, and as with such frameworks, balancing multiple objectives is inherently difficult. In particular, the coefficients $\lambda$ in Eq. (13) likely require careful tuning, which limits the scalability and robustness of the proposed method.
- The experiments are relatively limited. The authors should:
    1. Validate the effectiveness of the method on larger and more diverse base models, even without direct RL comparisons, simply demonstrating consistent performance gains would help.
    2. Provide more experimental details, such as the prompt templates, Avg@32 results, and the evaluation framework, as these directly affect the fairness and reproducibility of the evaluation [2].

Overall, I believe the paper does not yet meet the acceptance standard now. However, I also acknowledge that Dynamic RL, even without external prompts and answers, achieves significant improvements on simpler benchmarks and performs comparably to RL. Therefore, I recommend a borderline reject, while looking forward to the authors' response.

[1] Liu, Zichen, et al. "Understanding r1-zero-like training: A critical perspective." arXiv preprint arXiv:2503.20783 (2025).

[2] Hochlehnert, Andreas, et al. "A sober look at progress in language model reasoning: Pitfalls and paths to reproducibility." arXiv preprint arXiv:2504.07086 (2025).

**Questions:**

- I suspect that allowing the LLM to generate its own problems may largely exploit overlap with data the model has already seen during pretraining. If this is the case, Dynamic RL is effectively performing RL on existing problems rather than on genuinely novel ones. I would like to see the authors’ perspective on this point.

---

> ### Author Response · Authors · 2025-11-20
>
> We appreciate your careful and constructive comments. We have addressed the questions that you raised as follows. Please let us know if you have any further concerns.
>
> $\textbf{Q1:}$ The proposed method essentially leverages the LLM’s own self-consistency to provide learning signals for both its solver and question-setter roles.
>
> $\textbf{A1:}$ We agree that majority-voted estimation may fail to recover the golden answers. In fact, this is precisely the issue our paper aims to address. We discuss the limitations of this estimation method in Table 3 (Page 12). The issue you mentioned corresponds to the category “Underestimated Difficulty” in Table 3: the generated questions are too difficult for the method to reliably estimate golden answers.
>
> More importantly, this limitation is not specific to the majority-voted method. It is a general and unavoidable issue faced by all estimation methods: as question difficulty increases, recovering golden answers becomes inherently harder. We chose majority-voting because its properties make analysis more transparent. Moreover, [1] shows that many estimation methods are equivalent. We have also tried the entropy minimization method, which achieves similar performance as the majority-voted method.
>
> Our strategy for mitigating this issue is not to design a new estimation method, but rather to generate relatively more questions that the estimation method can reliably handle, such that their overall influence on training remains positive. Specifically:
>
> 1. Question reward design. We construct a question reward function that assigns higher rewards to questions with relatively more consistent answers.
>
> 2. Filtering unsuitable questions. We incorporate heuristic rules to remove questions that are unlikely to be estimated correctly.
>
> 3. Difficulty-controlled evolution. As shown in Section 3.4 (Figure 3), we regulate the evolution of question difficulty via $\lambda_q$. As the model improves, we gradually increase question difficulty, but never too rapidly; this preserves a sufficient proportion of questions that the estimation method can handle reliably. Training on such questions reinforces a improvement cycle. Overall, the difficulty of questions should align with the capabilities of both the current LLM and the estimation method.
>
> [1] has shown that semi-dynamic RL without golden answers can sustain only a limited number of training steps before performance deteriorates. Since questions are drawn from a static distribution with uncontrolled difficulty, estimation errors accumulate, gradually degrading the model. In contrast, dynamic RL supports more than a thousand steps, and ultimately reaches a level comparable to training with golden answers.
>
> [1] Zhang Y, Zhang Z, Guan H, et al. No Free Lunch: Rethinking Internal Feedback for LLM Reasoning[J]. arXiv preprint arXiv:2506.17219, 2025.
>
> $\textbf{Q2}$ On AIME24 and AIME25, the proposed method achieves only limited gains. Under a well-designed prompt template, Qwen2.5-Math-1.5B base can already reach a score of 16.7 on AIME 2024 [2], which is higher than that achieved by Dynamic RL.
>
> $\textbf{A2:}$ The reported evaluation metric pass@8 in [2] is different from our reported evalutation metric pass@1, which is greater than pass@1. Thus, the claim “the proposed method achieves only limited gains” is not accurate. Our method achieves a consistent performance improvement compared to base models as shown in Table 1. Notably, on the most challenging AIME25 dataset, our approach achieves the best performance. These results demonstrate that our method can improve the reasoning ability rather than merely increasing its answer confidence.
>
> [2] Liu, Zichen, et al. "Understanding r1-zero-like training: A critical perspective." arXiv preprint arXiv:2503.20783 (2025).
>
> $\textbf{Q3:}$ The method resembles multi-objective RL, balancing multiple objectives is inherently difficult.
>
> $\textbf{A3:}$  Although Eq.(13) contains four objectives, in practice only two hyperparameters, $\lambda_q$ and $\lambda_{dq}$, require tuning. Since the final objective is invariant under a global scalar multiplier, we can fix one hyperparameter (such as $\lambda_s$) to 1. Moreover, answer diversity has minimal influence on the other objectives. Thus, we set $\lambda_{da}=1$ to remove another hyperparameter.
>
> The remaining two hyperparameters are essential for stable long-horizon training: $\lambda_{dq}$ prevents mode collapse, and $\lambda_q$ controls the evolution of question difficulty. Both are required for sustaining training over more than a thousand of steps. This reflects the inherent cost of training on data that is neither pre-collected nor labeled. We will explore further reducing the number of hyperparameters through adaptive methods.
>
> We also conduct a sensitivity analysis of $\lambda_q$ and $\lambda_{dq}$ as shown in the Table 5 and Table 6 (Page 20). It can be seen that the performance does not oscillate with changes in $\lambda_q$ and $\lambda_{dq}$.

---

> ### Author Response · Authors · 2025-11-20
>
> $\textbf{Q4:}$ Validate the effectiveness of the method on larger and more diverse base models, even without direct RL comparisons, simply demonstrating consistent performance gains would help.
>
> $\textbf{A4:}$ The base model used in our experiments, Qwen2.5-Math-7B, is already a relatively large and widely adopted model. Moreover, our method consistently yields improvements over the base model; please refer to our response A2 for details.
>
> Since our study requires training the base model on a large-scale dataset and for more than a thousand steps to assess scalability, the computational and runtime costs are considerable. Training larger models, such as the Qwen-32B series, under the same extended schedule would require several weeks and cannot be completed in a short timeframe.
>
> $\textbf{Q5:}$ Provide more experimental details, such as the prompt templates, Avg@32 results, and the evaluation framework.
>
> $\textbf{A5:}$ We use the standard Qwen template, “Please reason step by step, and put your final answer within \\boxed{}.” because the base models belong to Qwen series. The reported results in our paper are already Avg@16, meaning that each dataset is evaluated 16 times and the average pass@1 score (i.e., Avg@16) is reported. All evaluation-related hyperparameters (e.g., temperature, top-p) are provided in Section 3.1 under “Hyperparameter Settings.” We have further clarified these details in the revision.
>
> $\textbf{Q6:}$ I suspect that allowing the LLM to generate its own problems may largely exploit overlap with data the model has already seen during pretraining. If this is the case, Dynamic RL is effectively performing RL on existing problems rather than on genuinely novel ones. I would like to see the authors’ perspective on this point.
>
> $\textbf{A6:}$ We cannot provide a definitive answer, as it depends on how one defines or verifies that “the data already exists during pre-training.” For any generated question, a reliable method would be required to determine whether it appeared in the training data in order to draw a firm conclusion.
>
> In our view, Dynamic RL can generate new questions beyond those produced by the base model for two reasons. First, RL continuously updates the policy model’s distribution during training. As training progresses, the support set $\\{x|\pi(x)>0\\}$ evolves, and after many training steps, it differs from that of the base model. Second, we manually inspected the generated questions at each training step and observed that they gradually become more difficult over time. Examples illustrating this trend are provided in Table 4 (Page 14). These observations provide evidence that RL can generate questions beyond the scope of the base model.

---

### Official Review · Reviewer_vbRn · 2025-11-02

**Soundness:** 2
**Presentation:** 2
**Contribution:** 1
**Rating:** 2
**Confidence:** 4

**Summary:**

- This paper introduces Dynamic RL, a novel framework designed to enable an LLM to continually improve its performance without relying on any static datasets or verifiers.
- The proposed method uses the model to generating its own training data by sampling both questions and corresponding solutions from its current policy. For the reward signal, the proposed method uses the majority voting as the groud truth labels for answer generation and the function of pass rate for question generation.

**Strengths:**

- The motivation of this paper is clear: using LLM itself for iterative self-optimization without external datasets and verifiers.
- The paper is easy to follow.

**Weaknesses:**

- The paper is highly similar to the existing work [1][2] for proposer/generator co-evolving.
- The framework's foundation depends on the assumption that majority voting over the model's own outputs can serve as a reliable proxy for ground truth. This is a very strong assumption. Also, the first question that needs to be clarified for self-improvement is: where does the training motivation come from?
- This paper lacks the comparison of baseline approaches: (1) self-rewarding methods[3][4], which leverage LLM-as-a-Judge for self-improvement with iterative SFT/DPO/PPO; (2) Unsupervised RL methods[5][6], which use various entropy-based reward estimation for RL.
- This paper lacks the analysis of the co-evolution of question generation and answer generation (e.g., dynamics of training/evaluation metrics, special reasoning behaviors), and the reason why they are beneficial to each other.
- The proposed method introduces four different reward terms balanced by coefficients, in addition to the threshold. The paper notes that the balance between these is important, but does not provide a sensitivity analysis.

[1] Absolute Zero: Reinforced Self-play Reasoning with Zero Data

[2] Co-evolving llm coder and unit tester via reinforcement learning

[3] Self-Rewarding Language Models

[4] Self-Improving Alignment with LLM-as-a-Meta-Judge

[5] EMPO: Fully Unsupervised LLM Reasoning Incentivization

[6] Self-Rewarding Reinforcement Learning for LLM Reasoning

**Questions:**

Same as the weaknesses.

---

> ### Author Response · Authors · 2025-11-20
>
> We appreciate your careful and constructive comments. We have addressed the questions that you raised as follows. Please let us know if you have any further concerns.
>
> $\textbf{Q1:}$ The paper is highly similar to the existing work [1][2] for proposer/generator co-evolving.
>
> $\textbf{A1:}$ There are substantial differences between our method and the approaches in [1] and [2]. Some of these distinctions are already discussed in the Related Work section.
>
> As noted in the Related Work, prior methods [1][2] rely on an external verifier, whereas our dynamic RL framework directly estimates gold answers from the model’s own outputs. Methods in [1][2] operate in the code domain, where execution environments provide verification. In contrast, our work focuses on the math domain, where no such verifiable environment exists. Our setting requires neither gold answers nor external verifiers, making the task more challenging. Our contributions center on generating relatively more questions that can be reliably estimated by the estimation method and enabling long-horizon training (over one thousand steps) without performance degradation in the absence of gold answers.
>
> Beyond the conceptual distinctions, there are also several important methodological differences:
>
> 1. Estimation mechanism.
> We estimate gold answers using majority voting, a procedure that does not appear in [1][2].
>
> 2. Question reward design.
> We design a question reward function explicitly to assign questions with relatively high consistent answers a large reward. The designs in [1][2] differ substantially.
>
> 3. Evolution of question difficulty.
> Our method controls the evolution of question difficulty and focuses on maintaining a sufficient proportion of questions that remain estimable as difficulty increases. This aspect is not shown in [1][2].
>
> 4. Mode collapse in question generation.
> We identify a mode collapse phenomenon in question generation and introduce a diversity reward to address it. This issue is not examined in [1][2].
>
> These differences highlight that our method addresses a distinct setting and introduces several technical innovations that are not explored in prior work.
>
> $\textbf{Q2:}$ The framework's foundation depends on the assumption that majority voting over the model's own outputs can serve as a reliable proxy for ground truth.
>
> $\textbf{A2:}$ We agree that majority-voted estimation may fail to recover the golden answers. In fact, this is precisely the issue our paper aims to address. We have discussed the limitations of this estimation method in Table 3 (Page 12).
>
> More importantly, this limitation is not specific to the majority-vote method. It is a general and unavoidable issue faced by all estimation methods in the absence of golden answers: as question difficulty increases, recovering golden answers becomes inherently harder.
>
> Our strategy for mitigating this issue is not to design a new estimation method, but rather to generate relatively more questions that the estimation method can reliably handle, such that their overall influence on training remains positive. Specifically:
>
> 1. Question reward design. We construct a question reward function that assigns higher rewards to questions with more consistent answers.
>
> 2. Filtering unsuitable questions. We incorporate heuristic rules to remove questions that are unlikely to be estimated correctly.
>
> 3. Difficulty-controlled evolution. As shown in Section 3.4 (Figure 3), we regulate the evolution of question difficulty via the $\lambda_{q}$. As the model improves, we gradually increase question difficulty, but never too rapidly; this preserves a sufficient proportion of questions that the estimation method can handle reliably. Training on such questions reinforces a stable improvement cycle. Overall, the difficulty of questions must remain aligned with the capabilities of both the current LLM and the estimation method.
>
> $\textbf{Q3:}$ Where does the training motivation come from for self-improvement?
>
> $\textbf{A3:}$ We have already described the motivation for dynamic RL in the third paragraph of the Introduction. Semi-dynamic RL still relies on static datasets composed of human-written questions and human-labeled answers, which are both finite and expensive to obtain. Moreover, since the questions are sampled from a fixed distribution that does not evolve with the model, the proportion of questions that remain effective for training decreases over time. This mismatch between static data and a continually improving model limits the scalability of semi-dynamic approaches. Dynamic RL is proposed precisely to address this limitation.
>
> Another long-term motivation is that as LLMs or AI systems potentially surpass human capabilities, humans may no longer be able to produce sufficiently effective or challenging training questions. In such a scenario, LLMs must be able to generate their own training data to continue improving.

---

> ### Author Response · Authors · 2025-11-20
>
> $\textbf{Q4:}$ This paper lacks the comparison of baseline approaches [3][4][5][6].
>
> $\textbf{A4:}$ We have clearly stated in the Introduction that “The core design principle of our dynamic RL is to encourage the model to generate relatively more questions whose answers can be reliably estimated by the estimation method, rather than to devise a new estimation method.” We also highlight this distinction in the Related Work section: “These methods primarily focus on estimation strategies. In contrast, dynamic RL emphasizes generating questions that can be reliably assessed by estimation methods.”
>
> Therefore, our paper does not aim to propose new estimation methods. In fact, our approach is compatible with many existing estimation techniques. The majority-vote estimator used in our experiments can be replaced with alternative estimators. Prior work  [7] has shown that many such estimation methods are equivalent, and we have also tried the entropy minimization method, which achieve similar performance as the majority-voted method for dynamic RL.
>
> Our semi-dynamic RL baselines are considerably stronger than baselines that rely purely on estimation methods, because they are trained directly on datasets with human labeled gold answers. [7] has shown that semi-dynamic RL without gold answers can sustain only a limited number of training steps before performance deteriorates. Since questions are drawn from a static distribution with uncontrolled difficulty, estimation errors accumulate, gradually degrading the model and preventing continued training. In contrast, dynamic RL supports more than a thousand of steps, maintains stable performance, and ultimately reaches a level comparable to training with gold answers.
>
> [7] Zhang Y, Zhang Z, Guan H, et al. No Free Lunch: Rethinking Internal Feedback for LLM Reasoning[J]. arXiv preprint arXiv:2506.17219, 2025.
>
> $\textbf{Q5:}$ This paper lacks the analysis of the co-evolution of question generation and answer generation.
>
> $\textbf{A5:}$ We present the model performance in Figure 2, which illustrates how performance evolves throughout training. Figure 3 further reports the distribution of r(q), showing how the difficulty of generated questions changes over time. Due to space constraints, the remaining training dynamics are included in the appendix of the revised version (Pages 19).
>
> The answer diversity reward steadily increases and eventually remains close to 1, as satisfying answer diversity is relatively easy. The question diversity reward also increases during the early stages of training and then stabilizes in the range of approximately 0.7 to 0.8, which prevents both mode collapse and excessive exploration.
>
> The trajectories of the question reward and solution reward highlight the adversarial relationship between these two objectives. In the early phase of training (Step 0 to Step 50), the model performance improves rapidly, leading to a sharp increase in the solution reward, while the question reward decreases because overly simple questions receive low scores. As training progresses (Step 50 to Step 450), the generated questions gradually become more challenging, causing the solution reward to decline and the question reward to rise. After this phase, both rewards continue to fluctuate within a stable range, maintaining an effective balance between question difficulty and model solvability.
>
> $\textbf{Q6:}$ The proposed method introduces four different reward terms balanced by coefficients, in addition to the threshold. The paper notes that the balance between these is important, but does not provide a sensitivity analysis.
>
> $\textbf{A6:}$ The threshold $\tau$ can be easily set by a relatively large value to assign questions with high consistent answers a large reward. Although Eq. (13) contains four objectives, in practice only two hyperparameters, $\lambda_{q}$ and $\lambda_{dq}$, require tuning. Since the final objective is invariant under a global scalar multiplier, we can fix one hyperparameter (such as $\lambda_{s}$) to 1. Moreover, answer diversity is a necessary and typically easy-to-satisfy condition, and it has minimal influence on the other objectives. Therefore, we set $\lambda_{da}=1$ to remove an additional hyperparameter.
>
> The remaining two hyperparameters are essential for stable long-horizon training: $\lambda_{dq}$ prevents mode collapse, and $\lambda_{q}$ controls the evolution of question difficulty. As demonstrated in Section 3.4, both are required for sustaining training over more than a thousand of steps. This reflects the inherent cost of training on data that is neither pre-collected nor labeled. We will explore further reducing the number of hyperparameters through adaptive strategies.
>
> The sensitivity analysis of $\lambda_{q}$ and $\lambda_{dq}$ is shown in the Table 5 and Table 6 (Page 20). It can be seen that the performance does not oscillate with changes in $\lambda_{q}$ and $\lambda_{dq}$.

---

### Official Review · Reviewer_eVRR · 2025-11-03

**Soundness:** 4
**Presentation:** 3
**Contribution:** 3
**Rating:** 8
**Confidence:** 4

**Summary:**

The authors design a synthetic data generation algorithm that generates both prompts and responses using a self-reinforcement technique. They deploy the algorithm on a math question-answering task. They assume access to a function that can evaluate whether two answers are equivalent, which presumably can be well-approximated for math answering tasks.

The reward function for responses samples several solutions, and rewards agreement by counting the number of solutions equivalent to the majority vote. The question reward function attempts to avoid collapse by guiding the answers toward questions of intermediate difficulty. Given the sample of solutions, they measure the fraction of unique solutions. The question reward function is parameterized by a target fraction tau, where specifies where reward is maximized. It tapers linearly for smaller or larger fractions. Intuitively, this avoids collapse by avoiding questions that the model is good at answering (or, more accurately, confident at answering), as well as overly-difficult questions.

They also introduce a diversity reward. One can measure the syntactic similarity between two questions by measuring token similarity. One can also measure whether the answer produced by the questions are equivalent. These can be used to encode a diversity score for a given parameterization of the model.

They employ a few additional practical tricks: 1) filtering questions that are likely to produce degenerate answers, and 2) standardizing reward functions.

Finally, they balance the various rewards with hyperpameters, and optimize the model using standard policy gradient.

Their empirical results evaluate their technique on a number of math datasets using Qwen2.5-Math-1.5B and 7B. Their results compete with related work that use external datasets to guide prompting. In contrast, their approach uses no external data. They provide ablation studies for each of the reward functions. They also provide observational results from the view of exploration/exploitation.

**Strengths:**

Strength #1: The paper is very clear, and combines a collection of conceptually clean ideas into a practical algorithm. With some minor exceptions, all the implementation details are provided, and the approach seems readily implementable.

Strength #2: I found the main empirical results to be impressive, demonstrating the effectiveness of the method against relevant work that makes use of external data.

**Weaknesses:**

Weakness #1: The exploration/exploitation section is by far the weakest part of the paper, and seems tacked-on at the end. I think it needs heavy editing. What are the colors in Figure 3? I thought r(q) is the answer support size? I also thought r(q) was the dependent variable in Figure 3? If so, what does r(q) = m = 16 mean? Is A_s introduced elsewhere in the paper?

Weakness #2: I find the paper to be very complete in its description of the algorithm. However one missing detail is the matching function S_a. While I believe there are candidates for S_a in a math-answering task, can the authors explain how this was implemented in their experiments? The existence of such a function seems critical to making this approach work in this (and other) settings.

**Questions:**

Question: Since dynamic RL does not use any data at all, is there a way to use the datasets (MATH-7.5K, DAPO, DeepScaleR) to warm-start the models somehow? Maybe by finetuning the base model? I suggest this only to make a more favorable comparison between the fully and semi-dynamic approach.

Grammar nits:
“training LLMs by itself” is grammatically broken. So is “All is done by large language model itself.” I can’t tell whether these are intentional. Consider “training the LLM by itself” and “all is done by the LLM itself.”

"However, the policy model πθ constantly changed during training, while the data is sampled from a static distribution, which can not be adapted to the evolving policy model πθ, inherently limits the learning scalability."

“we first samples questions from the policy”

“Since generating questions may also lead the model to produce solution implicitly”

More major: I’m having trouble parsing the final paragraph in section 3.4. I don’t understand what either sentence means. Please re-write.

---

> ### Author Response · Authors · 2025-11-20
>
> We appreciate your careful and constructive comments. We have addressed the questions that you raised as follows. Please let us know if you have any further concerns.
>
> $\textbf{Q1:}$ The exploration/exploitation section is by far the weakest part of the paper, and seems tacked-on at the end. I think it needs heavy editing. What are the colors in Figure 3? I thought r(q) is the answer support size? I also thought r(q) was the dependent variable in Figure 3? If so, what does r(q) = m = 16 mean? Is A_s introduced elsewhere in the paper?
>
> $\textbf{A1:}$ We have rewritten the exploration–exploitation section in the revised manuscript to improve clarity. The quantity $r(q)$, defined in Eq. (5), denotes the frequency of the majority answer. The case $r(q) = m = 16$ indicates that all sampled answers for a question are equivalent. The term $A_{s}$, defined in Eq. (14), represents the advantage corresponding to the solution reward function. We have added explicit cross-references in the revision to improve readability.
>
> Figure 3 is intended to illustrate that dynamic RL can generate relatively more questions whose answers can be reliably estimated by the estimation method. We achieve this by controlling the difficulty level of questions within each batch. We use the number of questions satisfying $r(q) = m = 16$ as a proxy for batch difficulty (the red curve in Figure 3): a larger value indicates easier questions. By tuning the hyperparameter $\lambda_{q}$, we maintain this quantity at a relatively high level (right panel of Figure 3).
>
> The curves in Figure 3 show the counts of questions with $r(q) = i$ for $1 \leq i \leq 16$, reflecting the full distribution of $r(q)$ throughout training. Since it is difficult to derive meaningful conclusions from curves other than the one for $r(q) = 16$, we have intentionally omitted legends for the remaining lines.
>
> $\textbf{Q2:}$ I find the paper to be very complete in its description of the algorithm. However one missing detail is the matching function S_a. While I believe there are candidates for S_a in a math-answering task, can the authors explain how this was implemented in their experiments?
>
> $\textbf{A2:}$ We have added citations for the matching function $S_a$ in the revised manuscript. The function $S_a$ can be computed in two ways. First, a rule-based method can be used to determine the equivalence of two mathematical objects, as in DeepSeek-R1; this approach can be implemented using the "math_verify" library, and we adopt this implementation in our work. Second, equivalence can also be assessed by the LLM itself, following the approach used in Kimi [1] (see Page 6, Paragraph “Reward Modeling for Math”).
>
> [1] Team K, Du A, Gao B, et al. Kimi k1. 5: Scaling reinforcement learning with llms[J]. arXiv preprint arXiv:2501.12599, 2025.
>
> $\textbf{Q3:}$ Since dynamic RL does not use any data at all, is there a way to use the datasets (MATH-7.5K, DAPO, DeepScaleR) to warm-start the models somehow? Maybe by finetuning the base model? I suggest this only to make a more favorable comparison between the fully and semi-dynamic approach.
>
> $\textbf{A3:}$ Thank you for the suggestions. We believe that keeping either the number of training steps or the number of training samples fixed is important for a fair comparison; therefore, we do not warm-start the model using additional datasets. Combining LLM-generated data with human-labeled data may leverage the strengths of both sources, and this is a promising direction for future exploration. We also anticipate that improving question quality and refining the question evolution process could further enhance the performance of dynamic RL, potentially enabling it to surpass semi-dynamic RL.
>
> $\textbf{Q4:}$ The typos.
>
> $\textbf{A4:}$ The typos have been corrected in the revision.

---

### Author Response · Authors · 2025-11-27

Dear Reviewers,

Thank you very much for your time and efforts in reviewing our submission. We are writing to respectfully follow up on our submitted responses. As the discussion deadline is approaching, please let us know if any further clarification or supplementary information is needed. We sincerely appreciate your consideration.

Kind regards,

Authors

---

### Note · Authors · 2026-01-31

**Comment:**

I have read and agree with the venue's withdrawal policy on behalf of myself and my co-authors.

**Withdrawal Confirmation:**

I have read and agree with the venue's withdrawal policy on behalf of myself and my co-authors.

---

### Meta-Review · Area_Chair_hhUU · 2026-01-07

**Summary:**

This papers a dynamic RL algorithm, which  enables an LLM to continually improve its performance without relying on any static datasets or verifiers. This paper were reviewed by three reviewers, with one positive recommendation and three negative recommendations. The major concerns are lack of novelty, lack of comparison with existing method, inadequate validation. After carefully reading the comments and rebuttals, the AC believes that this paper is not ready to be published in ICLR in its current form.

**Reviewer Concerns:**

The concerns of reviewer #eVRR seem to be addressed, but the remaining concerns of other reviewers are still outstanding.

**Reviewer Scores:**

The reviewers are unlikely to change their scores.

---

### Decision · Program_Chairs · 2026-01-26

Reject